# Association between Regular Exercise and Self-Rated Health and Sleep Quality among Adults in Japan during the COVID-19 Pandemic

**DOI:** 10.3390/ijerph181910515

**Published:** 2021-10-07

**Authors:** Hyuma Makizako, Ryoji Kiyama, Daisaku Nishimoto, Ikuko Nishio, Tomomi Masumitsu, Yuriko Ikeda, Misako Hisamatsu, Sachiko Shimizu, Masami Mizuno, Mikiyo Wakamatsu, Naomi Inoue, Takayuki Tabira, Tadasu Ohshige, Ayako Yamashita, Satoshi Nagano

**Affiliations:** 1Department of Physical Therapy, School of Health Sciences, Faculty of Medicine, Kagoshima University, Sakuragaoka 8-35-1, Kagoshima 890-8544, Japan; kiyama@health.nop.kagoshima-u.ac.jp (R.K.); ohshige@health.nop.kagoshima-u.ac.jp (T.O.); naga@m2.kufm.kagoshima-u.ac.jp (S.N.); 2Department of Nursing, School of Health Sciences, Faculty of Medicine, Kagoshima University, Sakuragaoka 8-35-1, Kagoshima 890-8544, Japan; daisaku@health.nop.kagoshima-u.ac.jp (D.N.); ikuko@health.nop.kagoshima-u.ac.jp (I.N.); t-masu@health.nop.kagoshima-u.ac.jp (T.M.); m-hisa@health.nop.kagoshima-u.ac.jp (M.H.); shimizu@health.nop.kagoshima-u.ac.jp (S.S.); mizuno-ma@health.nop.kagoshima-u.ac.jp (M.M.); mikiwaka@health.nop.kagoshima-u.ac.jp (M.W.); midwifeb@health.nop.kagoshima-u.ac.jp (N.I.); aya-yama@health.nop.kagoshima-u.ac.jp (A.Y.); 3Department of Occupational Therapy, School of Health Sciences, Faculty of Medicine, Kagoshima University, Sakuragaoka 8-35-1, Kagoshima 890-8544, Japan; yuriko@health.nop.kagoshima-u.ac.jp (Y.I.); tabitaka@health.nop.kagoshima-u.ac.jp (T.T.)

**Keywords:** middle-aged adults, self-rated health, exercise, physical activity, COVID-19

## Abstract

Regular exercise may be associated with better self-rated health and sleep status. However, this correlation among various age groups, such as young, middle-aged, and older people, as well as during the COVID-19 pandemic, has not been examined. This study examined the correlation between regular exercise and self-rated health and sleep quality among adults in Japan during the COVID-19 pandemic. Data were collected using an online survey conducted between February 26 and 27, 2021. A total of 1410 adults in Japan (age range, 20–86 years) completed the online survey. Regular exercise was divided into: (1) more than 30 min of moderate exercise a day, (2) more than 2 days per week, and (3) continuous for 1 year or longer. Self-rated health and sleep quality were assessed using the Likert scale. After adjusting for multiple confounders, regular exercise was correlated with decreased poor self-rated health and poor sleep quality in middle-aged adults; however, no significant correlation was observed among young and older adults. The promotion of regular exercise among middle-aged people during the COVID-19 pandemic may contribute to better self-rated health and sleep quality status.

## 1. Introduction

COVID-19 continued to be a global issue in 2021 [1]. The COVID-19 pandemic caused changes in lifestyle and health behavior, such as physical inactivity and sedentary behavior, among people of all ages, including university students, middle-aged adults, and older adults [2,3,4]. Sustained physical inactivity due to the COVID-19 pandemic has a negative impact on future health outcomes. In addition, physical activity may be strongly associated with a reduced risk of severe COVID-19 outcomes [5]. Thus, efforts to promote physical activity should be prioritized by public health agencies, even during the COVID-19 pandemic. Regular exercise under the COVID-19 pandemic among people of all ages could be key to a long-term healthy life.

In addition, a healthy lifestyle plays an important role in subjective outcomes. Self-rated health (SRH) is a good predictor of morbidity and mortality [6,7,8]. Regular exercise is associated with good SRH during normal social conditions [9]. In addition, poor sleep quality, which is a subjective health-related outcome, is associated with an increased risk of future diseases, such as metabolic syndrome [10], type 2 diabetes [11], coronary heart disease [12], and increased mortality [13]. In addition, there is a strong association between SRH and sleep quality among working-aged adults [14].

The COVID-19 pandemic situation may cause poor sleep quality. Young adults experienced high rates of sleep problems during the first two months (April to May 2020) of the pandemic [15]. During the COVID-19 quarantine, disruption of routine physical activity negatively influences sleep quality [16]. Therefore, promoting physical activity during the COVID-19 pandemic may be helpful in maintaining subjective health.

Regular exercise is associated with better SRH and sleep status [9,17]. However, this correlation among people in different age groups, such as young, middle-aged, and older, as well as during the COVID-19 pandemic, has not been analyzed to date. This study examined the correlation between regular exercise and SRH and sleep quality among adults in Japan during the COVID-19 pandemic.

## 2. Materials and Methods

### 2.1. Study Sample

Data for this study were collected from an online survey panel administered through the sampling of Y cloud systems among adults in Japan. The Y cloud system is a crowdsourcing service launched by Yahoo Japan Corporation, Inc. (Tokyo, Japan) in 2013. From 26 to 27 February 2021, 1602 adults in Japan aged 20 years and older completed the online survey. This online survey was conducted to establish a preliminary data platform as part of the Kagoshima University Online Health Laboratory (KU-OHL) project. Responders who reported a history of stroke, Parkinson’s disease, dementia, depression, and/or neurological disorders, or who gave the wrong answer to a question intended to identify fraudulent responses (choosing a specific option from multiple choices) were excluded. In addition, responders who did not identify their gender or education levels were excluded. Data from 1410 adults in Japan aged 20–86 years were analyzed.

### 2.2. Exercise Habits

Regular voluntary exercise was defined as having the following three criteria: (1) frequency of exercise of ≥2 times/week, (2) duration of ≥30 min per session, and (3) overall duration of ≥1 year [18]. Participants were asked to answer “yes” or “no” to the following question to identify regularly exercise habits: “Do you engage in moderate-intensity exercise (e.g., sweat lightly) at least twice a week for 30 min over a period of 1year?” Participants who answered “yes” to this question were classified as having regular exercise habits.

### 2.3. Self-Rated Health (SRH)

SRH was assessed based on the question: “How would you rate your general health status?” [19]. The responses were rated on a five-point Likert scale (“good,” “fairly good,” “Neither good nor poor,” “fairly poor,” and “poor”). Participants who answered “fairly poor” or “poor” were considered to have poor SRH.

### 2.4. Sleep Duration and Quality

Sleep duration (min/day) and quality were assessed using simple questions. To assess sleep duration, the participants were asked, “How long do you usually sleep?” [20]. Participants answered their usual sleep duration in ten-minute intervals. To assess sleep quality, the participants were asked: “How well do you sleep?” [21]. Sleep quality was assessed on a four-point Likert scale (“very well,” “fairly well,” “fairly poorly,” and “very poorly”), where “fairly poorly” or “very poorly” were considered to indicate poor sleep quality.

### 2.5. Demographic Variables

Age, gender, education, height, weight, living alone, living area (prefectures), and prescribed medication were recorded as demographic variables. Body mass index (BMI; kg/m^2^) was calculated based on height and weight. Ten prefectures in Japan, Tokyo, Kanagawa, Saitama, Chiba, Osaka, Hyogo, Kyoto, Aichi, Gifu, and Fukuoka, were identified as areas of special precautions based on the situation at the time of the survey (26 to 27 February 2021).

### 2.6. Statistical Analysis

Participant characteristics were expressed as means for continuous variables and population (%) for categorical variables. IBM SPSS version 25 (IBM Corp., Chicago, IL, USA) was used to perform all statistical analyses. The level of significance was set at *p* < 0.05. The difference in sample characteristics based on regular exercise habits was tested using Chi-squared tests for categorical variables and Student’s *t*-tests for continuous variables. A logistic regression model was used to examine associations between regular exercise and SRH and sleep quality. Adjusted models were used to examine correlations, including age, gender, education, BMI, living alone, living area, and medication (≥3 *n*/day) as covariates. Analyses were performed for the overall sample and for age categories, such as 20–39 years, 40–64 years, and 65 years and older.

## 3. Results

Data from 1410 adults (42.6% women) in Japan included in the final analyses are summarized in Table 1. The mean age of the participants was 44.6 ± 13.7 years. The 20–39 years, 40–64 years, and 65 years and older age groups contained 43.3% (*n* = 611), 44.0% (*n* = 621), and 12.6% (*n* = 178) of participants, respectively. Of the 1410 participants, 184 (13.0%) had poor SRH and 258 (18.3%) had poor sleep quality.

Overall, 391 patients (27.7%) had regular exercise habits during the COVID-19 pandemic. Comparisons of characteristics between participants with and without regular exercise habits are presented in Table 2. Participants with regular exercise habits were significantly older (*p* < 0.01), with a higher proportion of men (*p* < 0.01), and higher education levels (*p* = 0.04) than those without. There were significant differences in poor SRH between participants with regular exercise habits (8.2%) and those without (14.9%, *p* < 0.01). Although participants with regular exercise habits tended to have good sleep quality (*p* = 0.05), this difference was not significant. There was no significant difference in sleep duration between participants with regular exercise habits and those without.

Odds ratios (ORs) indicating associations between regular exercise habits and SRH and sleep quality are shown in Table 3 and Table 4, respectively. In logistic analyses using overall data, regular exercise was significantly correlated with lower rates of poor SRH (OR = 0.51, 95% confidence interval [CI] 0.34–0.77) after controlling for age, gender, education, BMI, living alone, living area, and medication (≥3 *n*/day). On the other hand, regular exercise was not significantly correlated with lower rates of poor sleep quality after adjusting for covariates (OR = 0.75, 95% CI 0.55–1.04).

In stratified analyses for age groups, after adjusting for age, gender, education, BMI, living alone, living area, and medication (≥3 *n*/day), a significant OR of regular exercise for poor SRH was observed among the middle-aged group, including adults of 40–64 years (OR = 0.41, 95%CI 0.21–0.79). There were no significant ORs among the young aged group, including adults of 20–39 years (OR = 0.67, 95%CI 0.35–1.30) and the old-aged group of 65 years and older (OR = 0.38, 95%CI 0.13–1.07). In addition, a significant OR of regular exercise (OR = 0.59, 95%CI 0.36–0.99) for poor sleep quality was observed after adjusting for covariates among the middle-aged group, while there were no significant OR of regular exercise among the young aged group, including adults of 20–39 years (OR = 0.84, 95%CI 0.51–1.37), and the old aged group (OR = 0.92, 95%CI 0.37–2.32).

## 4. Discussion

This study used online survey data of adults in Japan and indicated that regular exercise during the COVID-19 pandemic was associated with better SRH and sleep quality status. In particular, regular exercise was linked with lower rates of poor SRH and poor sleep quality in middle-aged adults. These results suggest that promotion of regular exercise among middle-aged people during the COVID-19 pandemic may contribute to better SRH and sleep quality.

Studies before the COVID-19 pandemic by the National Health and Nutrition Survey of Japan (NHNS-J 2012) showed that the majority of adults in Japan performed inadequate amounts of exercise, where 36% of men and 28% of women aged 20 years or older regularly exercised for at least 30 min two or more times a week [22]. According to a national survey conducted by the Ministry of Health, Labour and Welfare in Japan in 2019, the rate of regular exercise was 33.4% in men and 25.1% in women aged 20 years and older before the pandemic. Men aged 40–49 years (18.5%) and women aged 30–39 (9.4%) reported a lower rate of regular exercise [23]. Thus, there may be a need for middle-aged adults to be more health-conscious. The present study demonstrated that the proportion of regular moderate exercise of more than 30 min per day more than twice per week over a continuous period of one year or longer during the COVID-19 pandemic was 27.7%. Interestingly, a higher proportion of regular exercise (38.8%) was observed in older adults aged ≥65 years. Although several studies have indicated that the COVID-19 pandemic led to physical inactivity [24], some people may maintain exercise routines or reconsider their lifestyle to include regular exercise habits.

This study focused on only simple exercise habits, defined as moderate exercise lasting more than 30 min per day, more than twice per week, for a continuous period of at least 1 year. The study reported that Japanese women aged ≥20 years (25.1%) and older women aged ≥65 years (33.9%) exercised less regularly than men aged ≥20 years (33.4%) and older men aged ≥65 years (41.9%) [23]. Other questionnaires that assess the amount of spontaneous physical activity may provide information for health maintenance from different perspectives.

The proportion of Japanese people aged > 20 years (*n* = 2559) with poor sleep quality was 21.7% [25]. Poor sleep quality was found to be associated with poor perceived physical and mental health status [25]. In the present study, the proportions of poor SRH and poor sleep quality were 13.0% and 18.3%, respectively.

Based on strong evidence of regular physical activity for health benefits [26,27], guidelines for promoting physical activity are recommended. Regular exercise lasting over one year appears to be a key behavior to meet the recommended level of physical activity. Regularly engaging in physical activities is associated with perceived good health, including mental health [9,28,29,30]. A systematic review examining the interrelationship between sleep and exercise concluded that exercise improved sleep quality; however, no difference and a negative impact of exercise on sleep were observed [17]. Moreover, study results varied significantly due to the participants’ age. A previous systematic review showed moderately positive effects of an exercise training program on sleep quality in middle-aged adults [21]. The impact of regular exercise on sleep quality may differ depending on age group.

The present study indicated that regular exercise habits were associated with a lower rate of poor sleep quality in middle-aged adults; however this correlation was not observed in young and older adults. During the COVID-19 pandemic, lifestyles appeared to change in all age groups. Among middle-aged adults, many lifestyle aspects, such as work style, leisure time, and caring for children or acting as caregivers, changed. Lifestyles of middle-aged people changed dramatically due to the COVID-19 pandemic compared with younger and older age groups. Therefore, the impact of exercise routines on sleep quality and SRH may be significant. However, the impact of exercise was limited. Exercising alone and exercising with others may have health benefits; however, an increased frequency of exercise with others has important health benefits [31]. During the COVID-19 pandemic, people with regular exercise habits may exercise alone or with only familiar others. Exercise intensity and situation, such as being alone or with others, should be considered.

Several factors, such as health literacy, were indicated to encourage regular exercise in various populations, including children [32], university students [33], and community-dwelling older adults [34]. In particular, under unprecedented situations, such as the COVID-19 pandemic, higher health literacy could be helpful in maintaining or creating regularly exercise habits in daily life.

This study had several limitations. Regular exercise was self-reported and objective measures were not considered. Although the current study identified regular exercise as more than 30 min of moderate exercise per day, more than two days per week, and over a continuous period of one year or longer, details regarding exercise intensity and frequency were not obtained. The current survey did not include questions on changes in physical activity before and after the pandemic. Physical activity levels, including the intensity of exercise before the pandemic, were also not investigated. Changes in healthy lifestyle habits, such as those that are maintained, improved, or remain unchanged before and after the pandemic, may affect health status during the pandemic. In addition, participants in this study were recruited through an online survey and a non-random sampling method was used. Therefore, selection bias should be considered when interpreting the results. Other behavioral changes, such as work style, leisure time activity, and caring times for others due to the COVID-19 pandemic were not assessed. This study used simple questions to assess sleep quality and duration without objective measures. Therefore, answers may reflect not only sleep quality but also satisfaction and perception of sleep conditions. Future studies should use additional objective measures for sleep to verify these associations.

## 5. Conclusions

Regular exercise during the COVID-19 pandemic was linked with lower rates of poor SRH and poor sleep quality among middle-aged adults in Japan. However, these associations were not significant in young and older populations. These results suggest that promotion of regular exercise among middle-aged people during the COVID-19 pandemic may contribute to better SRH and sleep quality. Intervention strategies to maintain or create regularly exercise habits and clarify the intervention effects among middle-aged adults are required.

## Figures and Tables

**Table 1 ijerph-18-10515-t001:** Characteristics of participants.

Variable	Overall (*n* = 1410)	20–39 Years (*n* = 611)	40–64 Years (*n* = 621)	65 Years and Older (*n* = 178)
Age, years	44.6 ± 13.7	32.3 ± 5.4	49.5 ± 6.3	69.7 ± 4.2
Women, *n* (%)	600 (42.6)	344 (56.3)	202 (32.5)	54 (30.3)
Education, *n* (%)				
Master/doctorate degree	71 (5.0)	29 (4.7)	31 (5.0)	11 (6.2)
Bachelor’s degree	690 (48.9)	324 (53.0)	290 (46.7)	76 (42.7)
Professional degree	259 (18.4)	98 (16.0)	132 (21.3)	29 (16.3)
High school/junior high school graduate	390 (27.7)	160 (26.2)	168 (27.1)	62 (34.8)
Body mass index, kg/m^2^	22.1 ± 3.7	21.7 ± 3.8	22.5 ± 3.8	22.3 ± 2.9
Living alone, *n* (%)	275 (19.5)	129 (21.1)	126 (20.3)	20 (11.2)
Living aera (statement of COVID-19 emergency), *n* (%)	895 (63.5)	388 (63.5)	392 (63.1)	115 (64.6)
Medication (≥3 *n*/day), *n* (%)	140 (9.9)	26 (4.3)	57 (9.2)	57 (32.0)
Regularly exercise habits, *n* (%)	391 (27.7)	155 (25.4)	167 (26.9)	69 (38.8)
Self-rated health, *n* (%)				
Good	278 (19.7)	129 (21.1)	111 (17.9)	38 (21.3)
Fairly good	452 (32.1)	215 (35.2)	193 (31.1)	44 (24.7)
Neither good nor poor	496 (35.2)	197 (32.2)	231 (37.2)	68 (38.2)
Fairly poor	154 (10.9)	59 (9.7)	73 (11.8)	22 (12.4)
Poor	30 (2.1)	11 (1.8)	13 (2.1)	6 (3.4)
Sleep duration, min/day	405.1 ± 66.2	409.7 ± 68.4	394.7 ± 62.6	426.0 ± 64.0
Sleep quality, *n* (%)				
Very well	370 (26.2)	170 (27.8)	140 (22.5)	60 (33.7)
Fairly well	782 (55.5)	324 (53.0)	367 (59.1)	91 (51.1)
Fairly poorly	243 (17.2)	108 (17.7)	110 (17.7)	25 (14.0)
Very poorly	15 (1.1)	9 (1.5)	4 (0.6)	2 (1.1)

**Table 2 ijerph-18-10515-t002:** Comparisons of characteristics, self-rated health, sleep duration, and sleep quality between participants with regularly exercise habits and those without.

Variable	Regularly Exercise Habits (*n* = 391)	None or Irregularly Exercise Habits (*n* = 1019)	*p*
Age group, *n* (%)			
20–39 years	155 (39.6)	456 (44.7)	0.002
40–64 years	167 (42.4)	454 (44.6)	
65 years and older	69 (17.6)	109 (10.7)	
Women, *n* (%)	123 (31.5)	477 (46.8)	<0.001
Education (bachelor’s/master/doctorate degree), *n* (%)	228 (58.3)	533 (52.3)	0.043
Body mass index, kg/m^2^	22.0 ± 3.3	22.1 ± 3.9	0.495
Living alone, *n* (%)	78 (19.9)	197 (19.3)	0.794
Living aera (statement of COVID-19 emergency), *n* (%)	248 (63.4)	647 (63.5)	0.981
Medication (≥3 *n*/day), *n* (%)	37 (9.5)	103 (10.1)	0.717
Self-rated health, *n* (%)			
Good/Fairly good/Neither good nor poor	359 (91.8)	867 (85.1)	0.001
Fairly poor/Poor	32 (8.2)	152 (14.9)	
Sleep duration, min/day	409. 8 ± 63.6	403.3 ± 67.1	0.102
Sleep quality, *n* (%)			
Very well/Fairly well	332 (84.9)	820 (80.5)	0.054
Fairly poorly/Very poorly	59 (15.1)	199 (19.5)	

**Table 3 ijerph-18-10515-t003:** Poor self-rated health and regularly exercise habits.

	Crude, OR (95% CI)				Adjusted, OR (95% CI)			
	Overall	20–39 Years	40–64 Years	65– Years	Overall	20–39 Years	40–64 Years	65– Years
Regularly exercise habits	0.51 ** (0.34–0.76)	0.71 (0.38–1.31)	0.40 ** (0.21–0.74)	0.38 * (0.14–0.98)	0.51 ** (0.34–0.77)	0.67 (0.35–1.30)	0.41 ** (0.21–0.79)	0.38 (0.13–1.07)

Adjusted model: age, gender, education, body mass index, living alone, living area, and medication (≥3 *n*/day). * *p* < 0.05, ** *p* < 0.01.

**Table 4 ijerph-18-10515-t004:** Poor sleep quality and regularly exercise habits.

	Crude, OR (95% CI)				Adjusted, OR (95% CI)			
	Overall	20–39 Years	40–64 Years	65– Years	Overall	20–39 Years	40–64 Years	65– Years
Regularly exercise habits	0.73 (0.53–1.01)	0.91 (0.571.46)	0.60 * (0.36–0.99)	0.76 (0.32–1.80)	0.75 (0.55–1.04)	0.84 (0.51–1.37)	0.59 * (0.36–0.99)	0.92 (0.37–2.32)

Adjusted model: age, gender, education, body mass index, living alone, living area, and medication (≥3 *n*/day). * *p* < 0.05.

## Data Availability

There are no linked research datasets for this study. The authors do not have permission to share the data.

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
