# Peer review of "Association between Regular Exercise and Self-Rated Health and Sleep Quality among Adults in Japan during the COVID-19 Pandemic"

_ijerph, 2021, doi:10.3390/ijerph181910515_

Round 1
Reviewer 1 Report
Thank you very much for the opportunity to review the manuscript. It has been very interesting.
I would like to make a few comments with the intention of providing the best information to the readers:
What type of sampling was used?
Did the survey include questions on changes in physical activity before and after the pandemic? It would be interesting to know whether healthy lifestyle habits were maintained, improved or unchanged. I understand that you may not be able to introduce this in your manuscript. If you can, please report it.
In the discussion:
- Lines 165-167 can regular exercise in this age range be justified by scientific literature? It would be interesting to know if there is a deeper health awareness in this age range.
- Lines 174-175: the results are not justified, but repeated. Please look for an explanation.
Your manuscript contains very interesting information. However, more information could be reported, as you indicate in your limitations. I invite you to continue and extend your study.
Author Response
Reviewer 1
Thank you very much for the opportunity to review the manuscript. It has been very interesting. I would like to make a few comments with the intention of providing the best information to the readers:
Response
We appreciate your helpful and insightful comments. Please see our responses to your comments.
What type of sampling was used?
Response
The current study used data from Y cloud systems, which allowed interested registrants and applicants to participate in this study. Therefore, non-probability sampling (non-random sampling) was used in the current study. These points have been added to the section on limitations.
Location in the text
Line 226:
In addition, participants in this study were recruited through an online survey, and a non-random sampling method was used.
Did the survey include questions on changes in physical activity before and after the pandemic? It would be interesting to know whether healthy lifestyle habits were maintained, improved or unchanged. I understand that you may not be able to introduce this in your manuscript. If you can, please report it.
Response
Thank you for your helpful comment. Regrettably, the current survey did not include questions on changes in physical activity before and after the pandemic. I acknowledge that this is very important and may be an interesting one. This has been mentioned in the Discussion section.
Location in the text
Line 222:
The current survey did not include questions on changes in physical activity before and after the pandemic. Physical activity levels, including the intensity of exercise before the pandemic, were also not investigated. Changes in healthy lifestyle habits, such as those that are maintained, improved, or remain unchanged before and after the pandemic, may affect health status during the pandemic.
In the discussion:
- Lines 165-167 can regular exercise in this age range be justified by scientific literature? It would be interesting to know if there is a deeper health awareness in this age range.
Response
Thank you for your important insights and comments. This study used nearly identical questions for exercise habits as the current study. However, the rate of regular exercise is not presented for each age range. According to a national survey conducted by the Ministry of Health, Labor and Welfare in Japan in 2019, the rate of regular exercise was 33.4% for men and 25.1% for women aged 20 years and older before the pandemic. Men aged 40–49 years (18.5%) and women aged 30–39 years (9.4%) exercised less frequently (Ref1). Thus, there may be a need for middle-aged adults to be more health-conscious.
Ref 1. Ministry of Health, Labour and Welfare.
https://www.mhlw.go.jp/stf/seisakunitsuite/bunya/kenkou_iryou/kenkou/eiyou/r1-houkoku_00002.html
Location in the text
Line 166:
According to a national survey conducted by the Ministry of Health, Labour and Welfare in Japan in 2019, the rate of regular exercise was 33.4% in men and 25.1% in women aged 20 years and older before the pandemic. Men aged 40–49 years (18.5%) and women aged 30–39 (9.4%) reported a lower rate of regular exercise [23]. Thus, there may be a need for middle-aged adults to be more health-conscious.
- Lines 174-175: the results are not justified, but repeated. Please look for an explanation.
Response
Thank you for your comments. The paragraph has been simplified.
Location in the text
Line 185:
Data from the 1995 comprehensive survey by the Ministry of Health and Welfare, including more than 80 thousand people in Japan aged > 15 years, indicated that 9.8% of people reported poor SRH [24]. The proportion of Japanese people aged > 20 years (n = 2559) with poor sleep quality was 21.7 % [25]. Poor sleep quality was found to be associated with poor perceived physical and mental health status [25]. In the present study, the proportions of poor SRH and poor sleep quality were 13.0% and 18.3%, respectively.
Your manuscript contains very interesting information. However, more information could be reported, as you indicate in your limitations. I invite you to continue and extend your study.
Response
I appreciate you taking an interest in our study. Despite several limitations, sharing the current data with the readers of this journal could be beneficial.

Reviewer 2 Report
The study approches an important aspect of the lyfestle in the pandemic time. Some aspects of the detail of the intensity of the exerecise, in terms of spontaneous physical activity , especially before the time of the investigation, needs to be described
In addition the author should define and discuss the eventual differences in lifestyle among the different gender of the population studied .
For example the spontaneous physical activity amount , investigated by IPAQ questionnaire coulb be important .
Author Response
Reviewer 2
The study approaches an important aspect of the lifestyle in the pandemic time. Some aspects of the detail of the intensity of the exercise, in terms of spontaneous physical activity, especially before the time of the investigation, needs to be described.
Response
Thank you for your helpful comment. Unfortunately, the current survey did not include questions on spontaneous physical activity and the exercise intensity before the pandemic. I agree that this is a critical issue and could be intriguing to discuss. Accordingly, several sentences have been added to the Discussion section.
Location in the text
Line 222:
The current survey did not include questions on changes in physical activity before and after the pandemic. Physical activity levels, including the intensity of exercise before the pandemic, were also not investigated. Changes in healthy lifestyle habits, such as those that are maintained, improved, or remain unchanged before and after the pandemic, may affect health status during the pandemic.
In addition, the author should define and discuss the eventual differences in lifestyle among the different gender of the population studied. For example, the spontaneous physical activity amount, investigated by IPAQ questionnaire could be important.
Response
Thank you for your valuable insight. We understand its significance. Gender-based differences in lifestyle should also be discussed. This study focused on only simple exercise habits, such as moderate exercise lasting more than 30 minutes per day, more than twice per week, for at least 1 year or longer. The discussion section has been updated accordingly.
Location in the text
Line 178:
This study focused on only simple exercise habits, defined as moderate exercise lasting more than 30 minutes per day, more than twice per week, for a continuous period of at least 1 year. The study reported that Japanese women aged ≥20 years (25.1%) and older women aged ≥65 years (33.9%) exercised less regularly than men aged ≥20 years (33.4%) and older men aged ≥65 years (41.9%) [23]. Other questionnaires that assess the amount of spontaneous physical activity may provide information for health maintenance from different perspectives.

Reviewer 3 Report
Line 2: regular instead of regularly
Line 74 question concerning the exercise: consider rewriting
Line 79: quite good seems to be better than good and quite poor worse than poor. You may consider using fairly good and fairly poor
Line 223: regular instead of regularly
Author Response
Reviewer 3
Line 2: regular instead of regularly
Response
We have revised the term accordingly. Thank you for your suggestion.
Location in the text
Title
Association between regular exercise and self-rated health and sleep quality among adults in Japan during the COVID-19 pandemic
Line 74 question concerning the exercise: consider rewriting
Response
We rewrote the explanation for the exercise habits question.
Location in the text
Line 74:
“Do you engage in moderate-intensity exercise (e.g., sweat lightly) at least twice a week for 30 minutes over a period of 1year?”
Line 79: quite good seems to be better than good and quite poor worse than poor. You may consider using fairly good and fairly poor
Response
Thank you for your valuable insight. We used fairly good and fairly poor rather than quite good and quite poor, accordingly.
Location in the text
Line 79:
The responses were rated on a five-point Likert scale (“good,” “fairly good,” “Neither good nor poor,” “fairly poor,” and “poor”).
Line 80:
Participants who answered “fairly poor” or “poor” were considered to have poor SRH.
Table 1:
Fairly good
Fairly poor
Table 2:
Good/Fairly good/Neither good nor poor
Fairly poor/Poor
Line 223: regular instead of regularly
Response
Thank you for your suggestion.
Location in the text
Line 236:
Regular exercise during the COVID-19 pandemic was linked with lower rates of poor SRH and poor sleep quality among middle-aged adults in Japan.

Round 2
Reviewer 1 Report
Thank you very much for responding to the comments raised in the review of your manuscript.